# Microwave Passive Direction-Finding Method Based on NV Color Center in Diamond

**DOI:** 10.3390/mi14040774

**Published:** 2023-03-30

**Authors:** Qi Wang, Yusong Liu, Yue Qin, Hao Guo, Jun Tang

**Affiliations:** Key Laboratory of Instrumentation Science and Dynamic Measurement, School of Instrument and Electronics, North University of China, Taiyuan 030051, China; wangqi980720@163.com (Q.W.);

**Keywords:** nitrogen-vacancy color center in diamond, microwave passive direction-finding technology, microwave intensity measurement, weighted global least squares

## Abstract

In this study, we established a passive direction-finding scheme based on microwave power measurement: Microwave intensity was detected using microwave-frequency proportion integration differentiation control and coherent population oscillation effect converting the change in microwave resonance peak intensity into a shift of the microwave frequency spectrum, for which the minimum microwave intensity resolution was −20 dBm. The direction angle of the microwave source was calculated using the weighted global least squares method of microwave field distribution. This lay in the 12~26 dBm microwave emission intensity range, and the measurement position was in the range of (−15°~15°). The average angle error of the angle measurement was 0.24°, and the maximum angle error was 0.48°. In this study, we established a microwave passive direction-finding scheme based on quantum precision sensing, which measures the microwave frequency, intensity, and angle in a small space and has a simple system structure, small equipment size, and low system power consumption. In this study, we provide a basis for the future application of quantum sensors in microwave direction measurements.

## 1. Introduction

In recent years, quantum precision measurement and sensing technology have rapidly developed. The nitrogen-vacancy (NV) color center in the diamond is currently considered to be an excellent solid-state spin structure and has been widely applied in contexts of magnetic fields [1], electric fields [2], and temperature [3], etc. Based on optical detection magnetic resonance (ODMR), spin-based sensors have been developed, such as magnetometers [4], pressure sensors [5], inertial sensors [6], etc.

Passive direction-finding technology is a method used to detect target information and its spatial properties with a certain degree of accuracy based on existing electromagnetic wave information in space [7]. Because it is highly concealed, and thus does not emit microwaves to the surrounding area, this technology has been widely used in military and civilian applications, including the positioning of civil equipment, such as unmanned aerial vehicle and pseudo-base stations and multi-platform passive detection [8]. It can be used for target measurements in long-distance and complex environments. However, it often encounters challenges such as large equipment size, high power consumption, complex systems, and high requirements for signal processing capability [9,10].

In this paper, based on the characteristics of a diamond NV color center and microwave field distribution, we used a magnetic field intensity scan, continuous microwave excitation ODMR, and microwave-frequency proportion integration differentiation (PID) control and locking in order to convert spatial microwave intensity and frequency information into fluorescence information and thereby measure the characteristic parameters of microwaves. Through the measurement of microwave field distribution intensity, the unknown microwave source direction can be measured using the weighted global least squares method for the field distribution [11]. This paper provides a microwave source direction measurement scheme with simple system structure, small equipment size, low system power consumption and low requirements for system information processing capability.

## 2. Materials and Methods

An NV color center in the diamond is a combination of a nitrogen atom, which replaces a carbon atom, with its adjacent vacancy. Additionally, it is a stable luminous point defect with a $C_{3V}$ type symmetry; its crystal structure is shown in Figure 1a [12]. NV color center is considered to be a solid-state quantum sensor with a great performance. Its fluorescence property and electron spin property are the subject of research. Its energy level structure is shown in Figure 1b [13]. Electrons in an NV color center have spin triplet states in both the ground state A23 and excited state E3 (i.e., |0>,|+1>,|−1> state), a magnetic dipole transition of a magnetically tunable spin level, and a zero field splitting D = 2.87 GHz between the |0> and |±1> states. The electrons in the NV color center are in a degenerate |+1> state without the magnetic field; when the magnetic field acts, its |+1> state has zero field splitting [14]. When the microwave acts on the electron in the NV color center and resonates with the energy level from the |0> state to |±1> state, its quantum state is reversed. Due to the metastable state of the non-radiative transition A11↔E1, the excited fluorescence intensity of the diamond decreases and shows a trough in the ODMR signal [15].

The structure diagram of the space microwave field detection system designed in this paper in shown in Figure 2a. In this paper, a type Ib diamond synthesized at high temperature and high pressure was used. Its volume was 1.5 mm×1.5 mm×1.5 mm. After 4 h of electron irradiation and 2.5 h of high-temperature annealing, its nitrogen content was less than 200 ppm. A laser (CRYSTALASER, Reno, Nevada, USA| CL532-100-SO|50 mW 532 nm) was used to irradiate the diamond, and an electromagnetic coil was used to apply a magnetic field along the axis of diamond <111> crystals. A continuous microwave was applied to the diamond using a microwave source (KEYSIGHT, Santa Rosa, CA, USA| N5183B) to make the NV color center in the diamond transition to the state of |±1>. At the same time, an Olympus flat-field achromatic objective (THORLABS, Newton, NJ, USA| RMS10X) was used 10 times to measure the fluorescence excited from the diamond, and then a photoelectric detector (THORLABS, Newton, NJ, USA| PDA100A2) was used to collect the fluorescence, which is filtered by the 600~800 nm lens group (THORLABS, Newton, NJ, USA| FELH0600&FESH0800) and converged by a convex focusing lens. In this paper, an array antenna was used as the analog measurement microwave emission equipment, and its structure and simulation radiation field diagram are shown in Figure 2b,c. The array antenna is composed of eight subunit antennas. The main radiation direction Z is the physical centerline perpendicular to the surface of the microwave antenna; the X direction represents the short side of the array microwave antenna, and the Y direction represents the long side of the array microwave antenna. From the simulation radiation diagram, it can be seen that the microwave power gain of this antenna in the main radiation direction is 18.15 dB. At the same time, the port reflection coefficient of the antenna is shown in Figure 2d. It can be seen that the main radiation frequency of the array microwave antenna is approximately 2.9 GHz. Therefore, this article only needs to provide a magnetic field of mT magnitude for the diamond NV color center to cause a decrease in the resonant microwave frequency within the range of the ODMR spectral peak. At the same time, the ODMR spectral peak contrast of the diamond NV color center is conveniently kept within an ideal range.

Based on the two-channel microwave coherent population oscillation (CPO) effect of a diamond NV color center, this paper designs an intensity detection scheme for an unknown microwave. When the pump source with an angular frequency of ω and the detection source with an angular frequency of ω+φ are emitted into an active medium at the same time, and φ of them is less than or equal to the particle lifetime, the pump source can effectively scatter the transient modulated ground-state particles onto the detection light, thus limiting the absorption of the detection source energy and causing its spectrum to change [16]. We set the microwave frequency to be measured as 2.9063 GHz, which is transmitted by the array antenna described above. Based on the Zeeman splitting effect of diamond NV color center, the position of its ODMR spectral peak is regulated. The electromagnetic coil is used to apply a magnetic field along the <111> crystal axis direction of the diamond NV color center. The ODMR spectrum of the diamond NV color center is shown in Figure 3a. In order to obtain the best test results, a 1.5 mT magnetic field is applied. At the same time, in order to use a single peak in the ODMR spectrum and make the contrast of this peak as high as possible, a frequency modulation (FM) microwave is used as a detection source microwave [17] with a frequency of:(1)$$f_{mod}\left(t\right)=f_{LO}+f_{dev}\cos\left(2\pi f_{ref}\right)$$

In this formula, $f_{LO}$ is the FM-modulated carrier frequency, which is same as the microwave to be measured (2.9063 GHz); $f_{dev}$ is the FM modulation depth, which is 3 MHz; and $f_{ref}$ is the FM modulation frequency, which is 100 kHz. When the dual-channel microwave frequency resonates with the diamond NV energy level, the population of the NV color center changes, i.e., the corresponding position in the ODMR spectrum changes in intensity, which is called the spectral hole-burning phenomenon. The results are shown in Figure 3b. Under the same power of FM microwave emission intensity, the ODMR spectrum under the influence of different pump source microwave intensities also has different spectral hole-burning phenomena. That is, as the microwave emission intensity of the resonance source increases, the ODMR spectral signal amplitude of the diamond NV color center decreases, and the height of the peak formed by the resonance “hole burning” in the spectrum increases.

The ODMR spectral curve under the influence of the two-channel coherent population oscillation of the diamond NV color center is processed using a first-order differential operation, the results of which are shown in Figure 3c. The first-order differential operation used in this article is located in the field programmable gate array (FPGA) control system, where the upper and lower limits of the output amplitude of the first-order differential system were set, with the output amplitude limit of 7.28 V (−3.64–3.64 V). The horizontal line existing in Image A is the cutoff amplitude. This processing method transforms the change in the overall intensity and waveform of the ODMR spectral curve caused by the dual-channel coherent population oscillation into the change in the zero position on its first-order differential curve [18]. At the same time, the method of microwave closed-loop frequency PID locking is used to lock the zero position on the first-order differential curve. The drift amount Δf of the zero position is input to the FPGA control board. When the microwave intensity of the resonance source changes, the spectral morphology of CPO also changes, affecting the morphology of its first-order differential curve. When the microwave intensity increases, the depth of the “hole burning” on the CPO spectral curve increases, and the corresponding zero point on the first-order differential curve shifts to the left. This shift corresponds to a change in the microwave frequency of Δf. After calculation and processing, the feedback amount is input to the detection microwave source, which is used to control the FM microwave carrier frequency and realize the correction of the zero position on the first-order guide curve [19]. At the same time, the feedback output from FPGA to the microwave source is taken as the result of the microwave intensity to be measured. The changes in the detection of the curve are shown in Figure 3d. The final output ΔV is used as the microwave intensity to be measured. Additionally, the structure diagram of the FPGA information solution system is shown in Figure 4a. This system includes a band-pass filter, a low-pass filter, a multiplier, an amplifier, a linear region amplification module, an information solving module, and a PID locking module. Through an interconnection with a computer, the microwave signal parameters and the execution of the PID locking program can be directly controlled using a host computer control system. The transfer function of the locking system is:(2)$$T\left(z\right)=\frac{G\left(z\right)C\left(z\right)}{1+G\left(z\right)C\left(z\right)}$$

In this formula, $C\left(z\right)=K_{p}\left(1+K_{I}\frac{1}{z}+K_{D}z\right)$ is the transfer function of the PID frequency closed-loop locking system, where $K_{p}$ is the system scale coefficient, $K_{I}$ is the integral coefficient, $K_{D}$ is the differential coefficient, and G(z) is the transfer function of an open-loop linear system:(3)$$G\left(z\right)=\frac{2CV_{0}}{\Delta v^{2}}f_{dev}H\left(z\right)$$

In this formula, *C* is the used NV color center ODMR spectral contrast, which is 6.2%; $V_{0}$ is the fluorescence intensity, which is 155 mV; Δv is the full width at half height of the ODMR spectrum, which is 2 MHz; H(z) is the transfer function of the first-order low-pass filter, which is $H\left(z\right)=\frac{\alpha z}{z+\alpha -1}$; $\alpha =\frac{2\pi \left(f_{c}/f_{s}\right)}{2\pi \left(f_{c}/f_{s}\right)+1}$, $f_{c}$ is the cut-off frequency of the low-pass filter; and $f_{s}$ is the sampling frequency of the low-pass filter. By adjusting the PID parameters, the system is in the optimal state. The $K_{p}$ parameter regulates the error parameter multiple, which can improve the system’s error correction rate; $K_{I}$ parameter control error time integration can improve system time sensitivity; and the first-order differential component of the $K_{D}$ parameter adjustment error can shorten the system regression stability time. Through actual testing and alignment, the optimal PID parameters of the system are determined as: $K_{p}=120,K_{I}=90$, $K_{D}=180$.

The dynamic measurement of the microwave intensity can be carried out. With this scheme, the microwave intensity–feedback curve relationship can be established by setting the microwave intensity to be measured as the scanning mode. Because the variation of the ODMR spectral hole-burning position in the NV color center under the microwave coherent population oscillation effect is proportional to the microwave intensity measured in mW, during data collection, the resonant microwave source used the microwave intensity measured in dBm, which has an exponential relationship with the microwave intensity measured in mW. Therefore, an exponential form was used to fit the microwave intensity curve. We used exponential fitting to fit the curve. The results are shown in Figure 4b, and the microwave intensity sensitivity in this article is 108.89 mV/mW. Moreover, in the detection method described herein, the minimum detectable microwave intensity is −20 dBm:(4)$$y=A_{1}\times e^{\left(\frac{-x}{t_{1}}\right)}+A_{2}\times e^{\left(\frac{-x}{t_{2}}\right)}+A_{3}\times e^{\left(\frac{-x}{t_{3}}\right)}+y_{0}$$

Here, $y_{0}=0.544$, $A_{1}=-2.029\times 10{\;}^{-20}$, $A_{2}=-2.336\times 10{\;}^{-5}$, $A_{3}=-0.6107$, $t_{1}=-0.594$, $t_{2}=-2.810$, $t_{3}=-67.725$.

## 3. Results

The array antenna described above was used as an analog measurement antenna. The microwave intensity W in the area was obtained by measuring the microwave field emitted by the antenna in the space range [20]:(5)W=(ρ,θ,P)

In this formula, θ is the included angle between the measurement position and the vertical line of the microwave transmitting antenna surface, which is set as (−15°~15°) in this paper. ρ is the linear distance between the measuring position and the central position of the microwave antenna, which is set as (100~150 cm). P is the output power of the microwave source used at the transmitter, which is set as (12~26 dBm) and packeted with an interval of 2 dBm. By using the above microwave intensity detection method, the microwave intensity field distribution is measured in the two-dimensional plane area in the main lobe direction of the antenna transmitting a microwave field (its horizontal height is the same as the horizontal height of the antenna physical center). The spatial distribution of the antenna microwave field is obtained by sampling the data in the spatial range, and the spatial position information of the dataset is improved via the third-order spline interpolation method [21]. The results are shown in Figure 5a, and the input intensity of this microwave data is 26 dBm.

Based on the collected and constructed microwave field spatial distribution data, by measuring the microwave intensity at the cross position of five points in space, and by combining the data of these points, the weighted global least squares method of field distribution is used to restore the microwave source direction [22]; the schematic diagram for this is shown in Figure 5b. In the case of known microwave antenna radiation, for a microwave field formed by the radiation of a microwave source in an unknown direction, any position can be selected as the center point, and one point can be measured every 90° around the center. A total of five positions of microwave field intensity were measured, and the measurement structure formed an observation matrix [23]. We used a microwave field intensity distribution matching method based on weighted global least squares to calculate the relationship between the measurement position and the relative angle of the microwave source.

Based on Gauss–Markov model, due to the unequal accuracy of the observation matrix and the coefficient matrix, the weighting method proposed by Schaffrin et al. [24] was used to modify its weighted estimation constraint criteria as follows:(6)$$e_{y}^{T}P_{y}e_{y}+vec\left(E_{A}^{T}\right)P_{A}vec\left(E_{A}\right)=min$$

In this formula, y is the observation matrix; P is the weight matrix based on the field intensity distribution matrix; e and E are random errors; and vec() is a matrix vectorization operator. At the same time, the construction objective function is:(7)$$\Phi =e_{y}^{T}Q_{y}^{-1}e_{y}+e_{A}^{T}Q_{A}^{-1}e_{A}+2\lambda^{T}\left[y-Ax-e_{y}+\left(x^{T}⊗I_{m}\right)e_{A}\right]$$

In this formula, Q is the cofactor matrix of the matrix vector; ⊗ is Kronecker product; λ is n×1 Lagrange multiplier vector; $I_{m}$ is m×m Identity matrix; and A is a full rank coefficient matrix based on P. The iterative expression of the Lagrange multiplier method used to solve its estimated quantity $\hat{x}$ is:(8)$$\hat{x}={\left(A^{T}{\left(Q_{y}+\left({\hat{x}}^{T}Q_{0}\hat{x}\right)Q_{x}\right)}^{-1}A-\hat{t}Q_{0}\right)}^{-1}A^{T}{\left(Q_{y}+\left({\hat{x}}^{T}Q_{0}\hat{x}\right)Q_{x}\right)}^{-1}y$$
(9)$$\hat{t}={\left(y-A\hat{x}\right)}^{T}{\left(Q_{y}+\left({\hat{x}}^{T}Q_{0}\hat{x}\right)Q_{x}\right)}^{-1}Q_{x}{\left(Q_{y}+\left({\hat{x}}^{T}Q_{0}\hat{x}\right)Q_{x}\right)}^{-1}\left(y-A\hat{x}\right)$$

The relative relationship between the direction of the microwave source and the measured center point can be calculated. When selecting a five-point cross coordinate, the difference between the final angle determination result and the actual result is caused by the different distance interval between the five points. By setting the interval distance between different five points, many random actual measurements and calculations, as well as the calculation of the average angle error result, are carried out. The distance interval is set as 2–11 cm (the interval is 1 cm), and the average angle error result is shown in Figure 5c. When the distance between the five points is 5 cm, the average angle error reaches the minimum value of 0.24°. Because the method for selecting test points in this paper is to sample the location of points within the microwave radiation field area, the excessively long sampling distance interval will lead to the loss of correlation between the obtained microwave field intensity distribution matrix, resulting in difficulties in determining the relative location of microwave sources. At the same time, when the sampling interval distance is too short, the microwave intensity distribution matrix will contain less information, and it may appear that the microwave intensity data within the same group are very close or even the same (depending on the minimum resolution of microwave intensity), causing difficulties in ultimately calculating the direction of the microwave source. Additionally, due to the power of different microwave emissions, there are differences in the intensity of the microwave field formed in the region, which also lead to different results of angle determination. Based on different microwave emission powers, the microwave source angle in this area is randomly measured and determined, and the final result is calculated, as shown in Figure 5d. The result was measured at a distance of 5 cm. The maximum angle recognition error is 0.48° and the minimum angle recognition error is 0.03° under different microwave transmission power. For the same microwave antenna, although there are differences in input microwave intensity, the microwave radiation field distribution results are similar (there are differences in intensity). For different input intensities, an important reason for the difference in angle calculation error is the system’s minimum microwave intensity resolution limit. When the input microwave intensity is low in the microwave radiation field formed within the region, the number of indistinguishable regions of microwave intensity for the system will increase. These regions, in which it is difficult to distinguish between intensity differences, directly affect the final measurement results. As shown in Figure 5d, the highest, average, and minimum angular error results for experimental groups with higher-input microwave intensity are better than those for groups with lower-input microwave intensity.

## 4. Discussion

In this study, a passive microwave direction-finding method based on a quantum sensor was designed and implemented to measure the direction of an unknown microwave source. Compared to traditional passive microwave direction-finding schemes based on electronic devices, diamond NV color centers are used as microwave-sensitive devices in this paper. Its advantage is that the diamond NV color center converts microwave features into NV color center fluorescence spectral features, which solves the problem of traditional electronic devices requiring high-performance information sampling and devices to process microwave information. Considering the advantages of the actual physical volume of the diamond, it can detect microwave information in a very small spatial area. Additionally, we used the CPO effect of a diamond NV color center to detect unknown microwave features, including frequency and intensity information. Considering the fluctuation of the diamond NV color center ODMR spectrum under the CPO effect, we conducted a first-order differential processing on its spectral curve, converting spectral CPO hole burning information to its first-order differential curve, increasing the stability of information extraction. A PID locking system was used to lock the zero position of the first-order differential curve, and a negative feedback microwave frequency control system was employed to dynamically measure the microwave intensity. We set the microwave source to the intensity scanning mode and established a relationship between its microwave intensity and feedback locking curve. An array antenna was used as a microwave transmitting device to simulate the direction-finding operation of a microwave source. By detecting the two-dimensional intensity distribution of the spatial radiation region of the array antenna, and by using the third order spline interpolation method, the distribution map of the microwave radiation intensity field in the spatial region of the microwave antenna was restored. Considering that the microwave field intensity information detected by the diamond NV color center did not contain vector information, it is necessary to perform microwave intensity detection at multiple locations within a spatial range. In this study, a five-point cross position detection in a two-dimensional plane was selected. Considering the problem of unequal accuracy between this information and the microwave radiation intensity field, a weighted global least square field intensity distribution matching algorithm was used to restore the relative direction of the microwave source to be measured. Experiments were conducted on the angle recognition results under different measurement parameters, and the average angle error was 0.24° at a distance of 5 cm.

The dynamic microwave detection method based on diamond NV color centers constructed in this study is helpful for various microwave feature detection requirements: by changing the locking conditions of the FPGA information processing system on the CPO curve, high-precision microwave frequency detection can also be achieved [25]. If a rotary directional receiving antenna is used as an auxiliary device, the detection of the microwave direction of arrival (DOA) can be completed at a fixed point [26]. The method described in this article provides a new idea for the application of new quantum sensor devices represented by diamond NV color centers in traditional measurement engineering fields.

## Figures and Tables

**Figure 1 micromachines-14-00774-f001:**
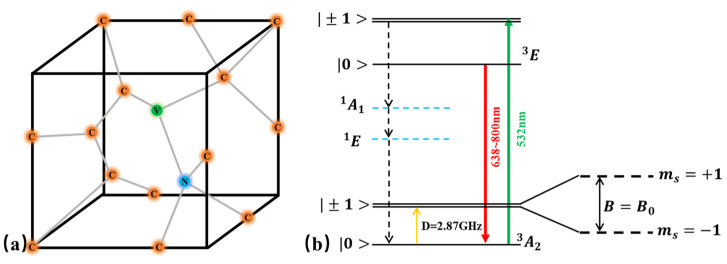
NV color center in diamond structure. (**a**) Schematic diagram of NV color center in diamond structure. (**b**) Energy level diagram of NV color center in diamond.

**Figure 2 micromachines-14-00774-f002:**
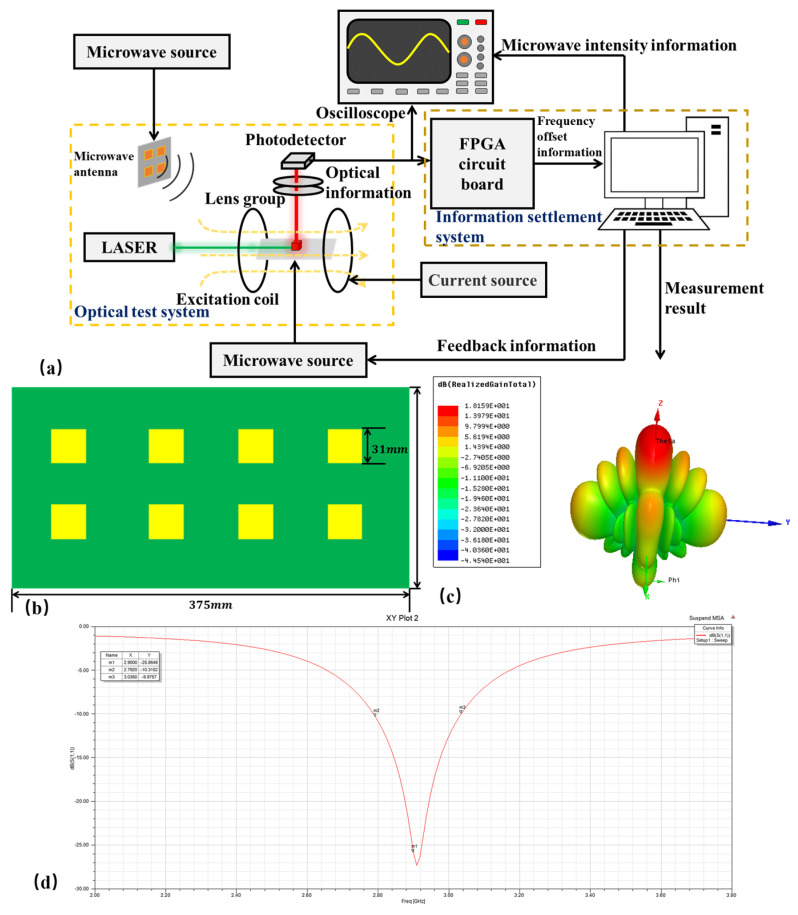
Schematic diagram of experimental system structure. (**a**) Schematic diagram of the overall structure of the experimental system. (**b**) Schematic diagram of array antenna structure. (**c**) Antenna radiation simulation diagram. (**d**) Simulated reflection coefficient of microwave antenna port.

**Figure 3 micromachines-14-00774-f003:**
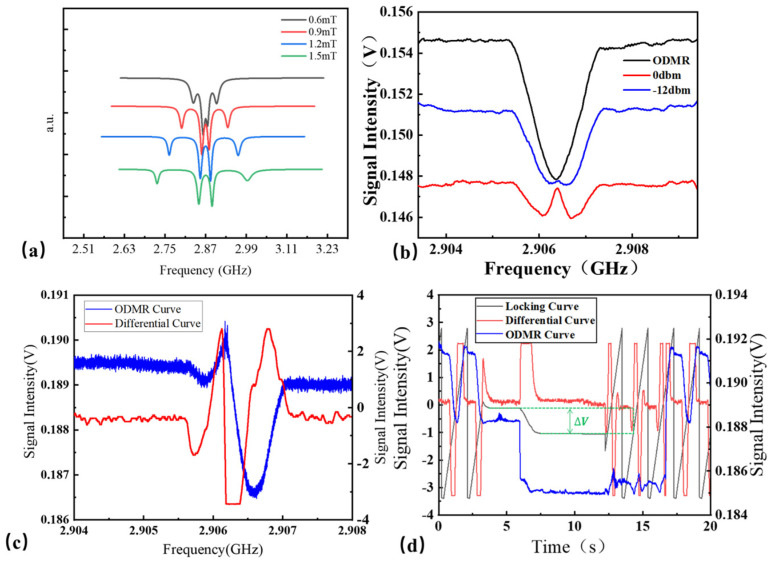
Diagram of experimental test results. (**a**) Schematic diagram of frequency detection method for fixed-frequency microwave using magnetic field intensity scan. (**b**) ODMR diagram of microwave frequency resonance with different intensities. (**c**) ODMR spectral curve in CPO and its first-order differential curve. (**d**) Microwave frequency closed-loop locking and microwave frequency resonance ODMR conversion diagram.

**Figure 4 micromachines-14-00774-f004:**
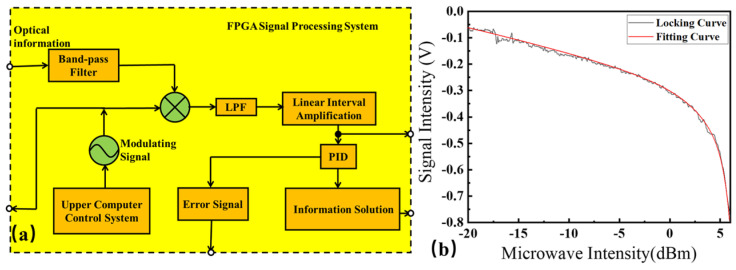
(**a**) Structure diagram of FPGA information processing system. (**b**) Microwave intensity locking curve and its fitting curve.

**Figure 5 micromachines-14-00774-f005:**
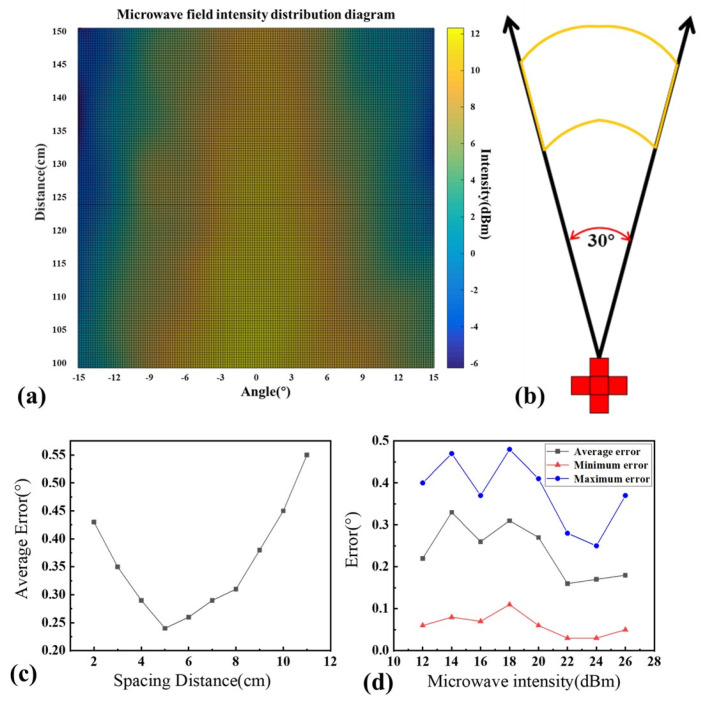
Test result diagram of microwave source direction. (**a**) Test results of antenna microwave field intensity distribution. (**b**) Microwave source direction measurement position and detectable range diagram. (**c**) The average error of the angle detection of the collected data at different distances of the five-point cross position. (**d**) Average error curve, minimum error curve, and maximum error curve of different microwave emission intensities at a distance of 5 cm.

## Data Availability

The data are available within this article.

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
