# Peer review of "Microwave Passive Direction-Finding Method Based on NV Color Center in Diamond"

_micromachines, 2023, doi:10.3390/mi14040774_

Round 1

Reviewer 1 Report

This paper proposes a microwave passive direction finding scheme based on diamond NV color centers as the detection core. Modern electromagnetic wave passive direction finding technology is of great importance. Compared to existing passive direction finding equipment, using diamond NV instead of electronic devices is innovative. It is suggested to publish after modification:

1.       The structure diagram of diamond NV color center in Figure 1(a) has some atomic position error, correct it please.

2.       The Figure 1(b) is blurred, replace with high-definition image please.

3.     Regarding the content described in Figure 4(b), please count the microwave intensity sensitivity in .

4.       Please mark the serial number for each formula in the article.

5.       On page 2, please correct the company name format in the device description:

KeySight N5183B > > > KEYSIGHT N5183B.

Reviewer 2 Report

The following modifications need be completed before publication:

1.      There are some images in the article that need to be replaced with a high-resolution version, such as: Figure 1(b), Figure 3(a).

2.      Some formulas are listed in this article, but they are all missing numbers, such as: Page 3, line 18.

3.     The description of  in line 14 on page 4 is relatively vague. The specific meaning of  should be explained in detail or marked in the image.

4.      The author declares that the red curve in Figure 3(c) is a first order differential curve, but there is an obvious horizontal line in it that does not match the first order differential of the ODMR curve. The author needs to explain this in detail.

5.      When establishing the relationship between the Locking Curve and microwave intensity, why does the author directly use exponential fitting without explaining the reason?
